# EFFICIENT SELF-GUIDED EDITING FOR TEXT-DRIVEN IMAGE-TO-IMAGE TRANSLATION

## ABSTRACT

Diffusion-based generative models achieve impressive text-driven image synthesis, largely due to classifier-free guidance (CFG), which enhances semantic alignment through blending conditional and unconditional denoising predictions. However, in text-guided image editing, CFG frequently induces structural drift, with the unconditional branch generating spatial mismatches. Prior approaches mitigate this by adding a reconstruction branch to steer the unconditional predictions, yet this consumes substantial GPU memory and computational resources. Our empirical studies uncover the inherent trade-off between semantic accuracy and structural integrity, pinpointing the null-text branch as the key culprit. We introduce a Target-Guided Unconditional Branch that repurposes semantic cues from the target prompt and initial denoising inputs from the source image to ensure spatial consistency. This method delivers superior editing quality without extra computational burden, serving as an efficient substitute for traditional CFG-dependent editing methods. Our experiments on PIE-Bench demonstrate that our method outperforms state-of-the-art baselines in structure preservation and background retention while maintaining comparable semantic alignment, all with reduced inference time and GPU memory usage.

## 1 INTRODUCTION

The recent advent of diffusion-based generative models Kim et al. (2022); Ramesh et al. (2022); Karras et al. (2022) has ushered in a new era of high-fidelity, text-driven image synthesis, led by models such as Stable Diffusion Rombach et al. (2022) and DALL-E Ramesh et al. (2022). These models build upon foundational advances in denoising diffusion probabilistic models (DDPMs) and their efficient variants like DDIM Sohl-Dickstein et al. (2015); Ho et al. (2020); Song et al. (2021). A key innovation enabling improved sample fidelity is classifier-free guidance (CFG) Ho & Salimans (2021), which blends predictions conditioned on a text prompt with those from a unconditional (null-text) prompt during the denoising process. This interpolation amplifies textual influence, and enhances semantic alignment with the prompt.

However, when these models are adapted from pure synthesis to the task of text-guided image editing, the very mechanism that strengthens semantic alignment can introduce significant structural drift. In particular, the unconditional branch of classifier-free guidance has been shown to accumulate spatial discrepancies, causing edited images to deviate from the original's layout and fine structural details Mokady et al. (2023); Tumanyan et al. (2023). As with DDIM inversion-based methods Song et al. (2021); Couairon et al. (2022); Roich et al. (2022); Balaji et al. (2022), which attempt to reverse the denoising path for editing, there exists an inherent tension between structure preservation and prompt alignment: low noise levels maintain geometry but reduce semantic responsiveness, while high guidance weights improve textual faithfulness at the expense of spatial fidelity Meng et al. (2021); Hertz et al. (2022); Crowson et al. (2022). Consequently, although classifier-free guidance enhances semantic fidelity, its application in editing contexts often results in outputs that adhere more closely to the target prompt's meaning but exhibit distortions in the original image's geometric structure and spatial coherence.

To elucidate this trade-off between structural fidelity and semantic alignment, we perform two systematic empirical studies (illustrated in Fig. 1). In Experiment 1, we compare editing with CFG enabled (unconditional branch active, 4th left row) against editing with CFG disabled (uncondi-

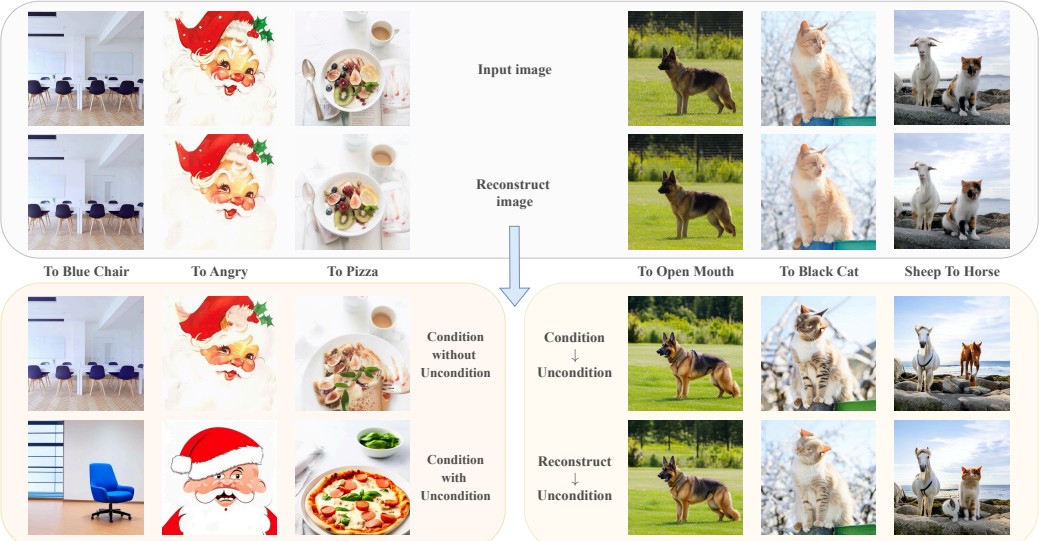

Figure 1: Illustration of the differences among the input images (1st row), reconstructed images (2nd row), edited images without classifier-free guidance (conditional-only, bottom-left 3rd row), edited images with classifier-free guidance (conditional and unconditional, bottom-left 4th row), edited images with self-attention injection from conditional to unconditional (bottom-right 3rd row), and edited images with self-attention injection from reconstruction to unconditional (PnP, bottom-right 4th row).

tional branch removed, 3rd left row). Our results indicate that disabling CFG preserves the input image's structure and scene composition but fails to effect meaningful semantic changes, whereas enabling CFG achieves strong semantic conformity at the expense of structural integrity as. In Experiment 2, we examine two distinct prompt-injection strategies: one that injects a source's branch into both the conditional and unconditional branches (4th right row), and another that injects only the conditional branch into the unconditional branch (3rd right row). We observe that selectively injecting the conditional branch into the unconditional branch yields the optimal balance, retaining the original image's spatial coherence while delivering substantive semantic edits.

These findings highlight the unconditional branch as the primary locus of structural degradation under CFG. Prior methods, such as Plug-and-Play (PnP) Tumanyan et al. (2023) and MasaCtrl Cao et al. (2023), mitigate this issue by introducing an auxiliary source-reconstruction branch that guides self-attention and feature injection into both the conditional and null branches. While effective, such approaches incur substantial additional memory and computational overhead, limiting their scalability in real time.

Inspired by recent advances in null-text (NT) inversion and direct inversion (DI), we instead propose a Target-Guided Unconditional Branch that obviates the need for a separate source-reconstruction pathway. Our approach reuses (i) feature and self-attention maps from the conditional branch to convey semantic intent, and (ii) the noised source image during early denoising steps to reinforce structural information. By modifying only the inputs and network of the unconditional branch, we achieve superior structure–semantics trade-offs without incurring extra GPU memory or inference time. Our experiments on PIE-Bench demonstrate that our method outperforms state-of-the-art baselines in structure preservation and background retention while maintaining comparable semantic alignment, all with reduced inference time and GPU memory usage.

## 2 RELATED WORK

**Diffusion Models.** Diffusion probabilistic models Liang et al. (2024); Luo et al. (2025); Shen et al. (2025) have emerged as a powerful class of generative models, leveraging iterative forward and reverse diffusion processes to model complex data distributions. Sohl-Dickstein et al. (2015)

introduced the foundational framework, inspired by non-equilibrium statistical physics, which systematically destroys and reconstructs data structures for flexible, tractable modeling. Building on this, Denoising Diffusion Probabilistic Models (DDPMs) Ho et al. (2020) achieve state-of-the-art image synthesis by combining variational bounds with denoising score matching, enabling high-quality generation and progressive lossy decompression. To address DDPM sampling inefficiency, Denoising Diffusion Implicit Models (DDIMs) Song et al. (2021) introduce a non-Markovian formulation that accelerates sampling while remaining compatible with DDPM training. In the domain of text-to-image generation, Latent Diffusion Models (LDMs) Rombach et al. (2022) operate in the latent space of pre-trained autoencoders to significantly reduce computational requirements while achieving state-of-the-art synthesis quality. By incorporating cross-attention mechanisms, LDMs enable flexible conditioning inputs like text and bounding boxes, achieving high-resolution, generation across tasks such as inpainting and super-resolution. unCLIP Ramesh et al. (2022) proposes a two-stage framework leveraging the joint text-image embedding with CLIP.

**Text-guided Image Editing.** Early editing methods Avrahami et al. (2021); Kwon & Ye (2021); Gal et al. (2022); Brooks et al. (2023) emphasized fine control over object attributes, layouts, and styles via fine-tuning model parameters or text embeddings. Notable examples include Imagic Kawar et al. (2023), UniTune Valevski et al. (2022), and SINE Zhang et al. (2023b), which allowed flexible changes to structures and visuals in single images. These relied on pre-trained generative models like diffusion frameworks, with sophisticated tools such as Imagen Saharia et al. (2022) and Stable Diffusion providing foundations for high-fidelity edits. GLIDE Nichol et al. (2021) pioneered diffusion-based text-conditional image synthesis. It compares classifier-free and CLIP-guided strategies, achieving better photorealism and caption alignment than DALL-E. GLIDE supports tasks like text-based inpainting. However, they often required heavy computational training, limiting flexibility and efficiency. In contrast, inversion-based techniques enable editing without altering model weights. Prompt-to-Prompt Hertz et al. (2022) accurately reconstructs sources and allows prompt-level semantics by optimizing unconditional embeddings per denoising step. Null-text Inversion Mokady et al. (2023) advances this with guided diffusion for improved reconstruction and control. Negative Miyake et al. (2025) further shows similar results without explicit null-text tuning. Recently, Direct Inversion Ju et al. (2023) offers an efficient solution for fast, high-quality inversion.

## 3 PRELIMINARY

**Self-Attention.** The self-attention mechanism Vaswani et al. (2017) captures long-range dependencies by allowing the model to capture interactions between different positions across various representation subspaces. Given a sequence of features $\Phi = [\phi_1, \cdots, \phi_N] \in \mathbb{R}^{N \times d}$, where $N$ represents the sequence length and $\phi_i \in \mathbb{R}^d$ denotes the $i$-th input feature of dimension $d$, the features $\Phi$ are first projected into queries ($Q$), keys ($K$), and values ($V$) using learnable weight matrices $W_q, W_k, W_v \in \mathbb{R}^{d \times d}$. The self-attention operation is defined as:

$$\text{Attn}(Q, K, V) = \text{Softmax} \underbrace{(QK^T/\sqrt{d})}_{A} V. \tag{1}$$

where the attention map $A \in \mathbb{R}^{N \times N}$ captures contextual relationships among input tokens. The Transformer leverages this mapping to aggregate information for each token based on all other tokens in the sequence. For simplicity, the term $\sqrt{d}$ is omitted in the self-attention computation throughout the remainder of the paper, as it is merely a scalar.

**Diffusion Models.** Diffusion models are generative frameworks that learn to generate data by reversing a gradual noise addition process. They model a Markov chain of diffusion steps that transform a data distribution into noise, then reverse it to produce new samples. Formally, for a data sample $\mathbf{x}_0 \sim p_{\text{data}}(\mathbf{x})$, the forward diffusion adds Gaussian noise over $T$ steps:

$$q(\mathbf{x}_t \mid \mathbf{x}_{t-1}) = \mathcal{N}(\mathbf{x}_t; \sqrt{1 - \beta_t}\mathbf{x}_{t-1}, \beta_t \mathbf{I}),$$

where $\beta_t$ is a predefined variance schedule controlling noise at timestep $t$. The reverse process synthesizes data from noise:

$$p_\theta(\mathbf{x}_{t-1} \mid \mathbf{x}_t) = \mathcal{N}(\mathbf{x}_{t-1}; \boldsymbol{\mu}_\theta(\mathbf{x}_t, t), \sigma_t^2 \mathbf{I}),$$

where $\boldsymbol{\mu}_\theta(\mathbf{x}_t, t)$ is a learned mean estimator, typically a neural network predicting the noise component $\boldsymbol{\epsilon}_\theta(\mathbf{x}_t, t)$.

**DDIM Inversion.** In this paper, variables with $\hat{\cdot}$ denote DDIM inversion, while those with $\cdot'$ refer to editing generation. Denoising Diffusion Implicit Models (DDIM) augment traditional diffusion frameworks. They incorporate a non-Markovian backward mechanism. This enables swifter sampling without diminishing output excellence. DDIM inversion serves as a method to revert a genuine picture $\mathbf{x}_0$ to a noisy variant $\mathbf{x}_T$ via the progressive noising procedure. The DDIM advancement phase is expressed as: $\mathbf{x}_t = \sqrt{\alpha_t}\mathbf{x}_0 + \sqrt{1-\alpha_t}\boldsymbol{\epsilon}$, where $\alpha_t = \prod_{s=1}^{t}(1-\beta_s)$ denotes the aggregated noise timetable, and $\boldsymbol{\epsilon}$ samples from the standard gaussian noise. The inversion phase is:

$$\hat{\mathbf{x}}_t = \frac{\sqrt{\alpha_t}}{\sqrt{\alpha_{t-1}}}\hat{\mathbf{x}}_{t-1} + \sqrt{\alpha_t}\left(\sqrt{\frac{1}{\alpha_t}-1} - \sqrt{\frac{1}{\alpha_{t-1}}-1}\right)\epsilon_\theta\left(\hat{\mathbf{x}}_{t-1}, t-1, y\right), \qquad (2)$$

which $y$ is the source prompt or just null. Leveraging the noise prediction model, DDIM inversion follows a deterministic trajectory, which ensures a unique mapping between input images and noise states ($\boldsymbol{x}_0 \to \hat{\boldsymbol{x}}_T$). Following the inversion, conditioned editing is performed by using a different condition $y_\mathcal{T}$ and $\epsilon_\theta(\boldsymbol{x}'_t, y_\mathcal{T}, t)$, with $\boldsymbol{x}'_T = \hat{\boldsymbol{x}}_T$, to generate an edited image $\boldsymbol{x}'_0$, i.e., $\hat{\boldsymbol{x}}_T \to \boldsymbol{x}'_0$. This inversion-generation paradigm enables controlled editing, which retains fidelity to the original images while allowing flexible modifications.

**Classifier-Free Guidance.** Classifier-Free Guidance improves the standard of outputs from text-guided diffusion models by harmonizing conditioned and unconditioned creation. In the sampling phase, classifier-free guidance merges the conditioned estimate (directed by a text cue $y$) with an unconditioned estimate. The adjusted noise forecast is expressed as:

$$\tilde{\epsilon}'_\theta(\mathbf{x}'_t, t, y) = \omega \cdot \underbrace{\boldsymbol{\epsilon}_\theta(\mathbf{x}'_t, t, y)}_{\text{Condition}} + (1-\omega) \cdot \underbrace{\boldsymbol{\epsilon}_\theta(\mathbf{x}'_t, t, \emptyset)}_{\text{Uncondition}}, \qquad (3)$$

where $\epsilon_\theta(\mathbf{x}'_t, t, y)$ represents the noise estimated using the text cue $y$, $\epsilon_\theta(\mathbf{x}'_t, t, \emptyset)$ is the noise estimated with an empty cue, and $\omega \geq 1$ serves as the guidance factor regulating the intensity of the text guidance. This technique strengthens the impact of the desired cue but creates imbalance when the reversal procedure employs an empty or origin cue, inspiring our suggested approach to apply the desired cue in the reversal. In our paper, to ensure clear differentiation among these two areas, this article employs $\mathcal{T}$, and $\emptyset$ to denote the target, and empty text, respectively.

## 4 METHOD

We now introduce *Efficient Self-Guided Editing (ESG)*, a framework designed to improve structure preservation and semantic fidelity in DDIM-based text-driven image editing, without requiring an additional source-reconstruction branch. ESG addresses the primary source of structural degradation in classifier-free guidance—namely, the unconditional branch—by reusing intermediate representations from the target-guided branch and modifying its input via source noise sharing, all without extra overhead.

### 4.1 TARGET-GUIDED UNCONDITIONAL BRANCH

Our empirical findings in Introduction and previous research Mokady et al. (2023); Ju et al. (2023) show that the unconditional branch is a major contributor to structural drift when classifier-free guidance is enabled. We propose injecting features from the conditional (target-prompt) branch into the unconditional (null-text) branch during the U-Net forward pass as shown in the circled 1 Figure 2. This injection promotes structural alignment and reduces semantic inconsistencies between branches. Inspired by evidence that mid-level layers in diffusion U-Nets Ronneberger et al. (2015); O'shea & Nash (2015); He et al. (2016) encode spatial layout and high-level semantics Tumanyan et al. (2023); Esser et al. (2021), ESG selects these layers and replaces the hidden states in the unconditional branch with those from the conditional branch.

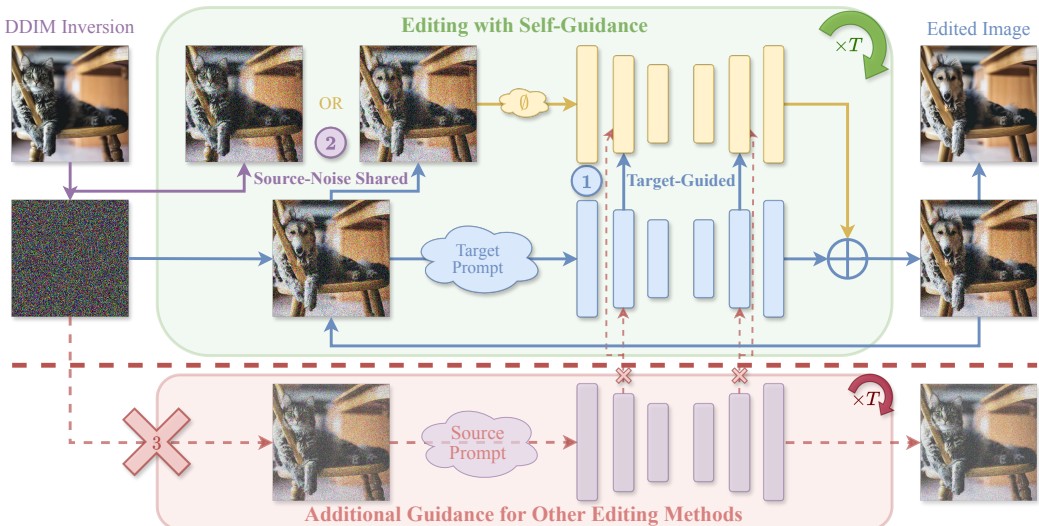

Figure 2: Illustration of our proposed Editing with Self-Guidance (ESG) framework for text-driven image-to-image translation at timestep $t$. The framework transforms an input image (upper-left) into a target-aligned image (upper-right). The left side (purple) depicts DDIM inversion, which converts the input image to initial latent noise. The upper-middle (green) outlines ESG process: the blue path represents the conditional branch with the target prompt, while the yellow path is the unconditional branch with null text. We inject features and self-attention maps from the target (conditional) branch into the unconditional branch (target-guiding, circled **1**). Additionally, we use the noised source image as input to the unconditional branch for $t > \tau$ to preserve shape and background (source noise sharing, circled **2**). The lower-middle (red) depicts the extra branch (circled **3**) needed by other methods, incurring additional computation and memory that ESG avoids.

**Convolutional Features and Self-Attention maps.** We define the injection operation for convolutional features and self-attention maps as:

$$f^{l,\emptyset} = f^{l,\mathcal{T}}(\mathbf{x}_t'^{,l}, \ t, \ y_{\mathcal{T}}), \tag{4}$$

$$\text{Attn}^{l,\emptyset} = \text{Softmax}(A^{l,\mathcal{T}})V^{l,\emptyset}, \tag{5}$$

where $f^{l,\emptyset}$ and $f^{l,\mathcal{T}}$ denote the $l$-th hidden features in the unconditional and conditional branches; $(Q^{l,\emptyset}, K^{l,\emptyset}, V^{l,\emptyset})$ are the query/key/value tensors for the unconditional branch; $(Q^{l,\mathcal{T}}, K^{l,\mathcal{T}}, V^{l,\mathcal{T}})$ are those for the conditional branch; and $A^{l,\mathcal{T}} = Q^{l,\mathcal{T}}(K^{l,\mathcal{T}})^T$. As shown in Figure 2 (circled **1**), this cross-branch injection enables the unconditional branch to leverage semantically aligned representations. It eliminates the need for an external source-reconstruction branch (circled **3** in Figure 2) and reduces guidance mismatch.

However, although feature injection enhances semantic consistency, it does not fully eliminate structural drift, particularly at high noise levels (large timestep $t$), where DDIM samples become increasingly unstable (as shown in the third row of Figure 1). In these early stages, features extracted from the noised image are often unreliable, leading to errors that accumulate during denoising.

## 4.2 SOURCE NOISE SHARED IN THE UNCONDITIONAL BRANCH

To reinforce structure preservation, ESG reuses the noised latent trajectory from DDIM inversion of the source image in the unconditional branch during early denoising steps. Unlike previous methods such as null-text inversion Mokady et al. (2023) or direct inversion Ju et al. (2023), which optimize the inversion process itself to improve reconstruction fidelity, our approach leaves the inversion unchanged and instead modifies the forward denoising process by sharing source noise selectively in the unconditional branch. As illustrated in Figure 2 (circled **2**), we define a timestep threshold $\tau$ that switches the unconditional input from the DDIM-inverted latent $\hat{\mathbf{x}}_t$ to the evolving editing state

$\mathbf{x}'_t$. Formally:

$$\epsilon_\theta^\emptyset = \begin{cases} \epsilon_\theta(\hat{\mathbf{x}}_t, t, \emptyset), & t > \tau \\ \epsilon_\theta(\mathbf{x}'_t, t, \emptyset), & t \le \tau, \end{cases} \tag{6}$$

where $\tau$ controls the transition between the inversion-based input and the edited latent input. Meanwhile, the target branch remains conditioned on $\mathbf{x}'_t$ and the prompt $y_\mathcal{T}$, i.e., $\epsilon_\theta^\mathcal{T} = \epsilon_\theta(\mathbf{x}'_t, t, y_\mathcal{T})$. where $\hat{\mathbf{x}}_t$ is the latent from the DDIM inversion of the source image, and $\mathbf{x}'_t$ is the current sample from the editing trajectory. The target branch remains conditioned on $y_\mathcal{T}$ at each step:

$$\epsilon_\theta^\mathcal{T} = \epsilon_\theta(\mathbf{x}'_t, t, y_\mathcal{T}).$$

Both $\hat{\mathbf{x}}_t$ and $\mathbf{x}'_t$ originate from the same inverted noise $\hat{\mathbf{x}}_T$, obtained by DDIM-inversion the source image $\mathbf{x}_0$. This shared origin ensures structural consistency across branches throughout the editing process.

The final CFG prediction is then computed as:

$$\tilde{\epsilon}'_\theta = \epsilon_\theta^\emptyset + \omega(\epsilon_\theta^\mathcal{T} - \epsilon_\theta^\emptyset), \tag{7}$$

where $\omega$ is the guidance scale. By anchoring the unconditional branch to the input image's shape while guiding semantics through the target branch, ESG achieves a more robust trade-off between structure and semantics.

### 4.3 EDTING WITH SELF-GUIDANCE

The complete ESG pipeline is outlined in Algorithm 1. All operations occur within a single CFG-based denoising trajectory. In each step, we reuse (1) convolutional features and attention maps from the target branch, and (2) source noise from the DDIM-inverted latent (for $t > \tau$). This design enables the unconditional branch to serve as a structure-preserving anchor while remaining semantically aligned via target-branch guidance. Specifically, it avoids adding new sampling paths or increasing memory overhead, unlike PnP or MasaCtrl. The resulting edited images maintain strong adherence to both the original image's layout and the semantics of the target prompt, demonstrating the effectiveness of self-guidance in a single-trajectory diffusion framework.

---

**Algorithm 1** Editing with Self-Guidance

**Input:** Source image $\mathbf{x}_0$, source prompt $y_\mathcal{S}$, target prompt $y_\mathcal{T}$, guidance scale $\omega$, switch timestep $\tau$, timestep set $\{T, \ldots, 0\}$
Initialize invested noise image list NL
**Stage A: DDIM Inversion**
**for** $t = 0$ **to** $T - 1$ **do**
    $\hat{x}_{t+1} \leftarrow$ DDIM-inversion$(\hat{x}_t, \hat{\epsilon}_t)$, and NL $\leftarrow \hat{x}_{t+1}$
**end for**
**Stage B: Single-trajectory Editing with Self-Guidance**
$\mathbf{x}'_T \leftarrow \hat{\mathbf{x}}_T$
**for** $t = T$ **to** $0$ **do**
    $\epsilon_\theta^\mathcal{T} = \epsilon_\theta(\mathbf{x}'_t, t, y_\mathcal{T})$
    **if** $t > \tau$ **then**
        $\mathbf{x}_t^\emptyset = \hat{\mathbf{x}}_t$
    **else**
        $\mathbf{x}_t^\emptyset = \mathbf{x}'_t$ as Eq. equation 6
    **end if**
    $\epsilon_\theta^\mathcal{T} = \epsilon_\theta(\mathbf{x}'_t, t, \mathcal{T}),$
    $\epsilon_\theta^\emptyset = \epsilon_\theta(\mathbf{x}_t^\emptyset, t, \emptyset)$ with $\begin{cases} f^{l,\emptyset} \leftarrow f^{l,\mathcal{T}} \text{ as Eq. equation 4} \\ A^{l,\emptyset} \leftarrow A^{l,\mathcal{T}} \text{ as Eq. equation 5} \end{cases}$
    $\tilde{\epsilon}'_\theta = \epsilon_\theta^\emptyset + \omega(\epsilon_\theta^\mathcal{T} - \epsilon_\theta^\emptyset)$ as Eq. equation 7
    $x'_{t-1} \leftarrow$ DDIM-sample$(x'_t, \tilde{\epsilon}'_\theta)$
**end for**
**return** $x'_0$

---

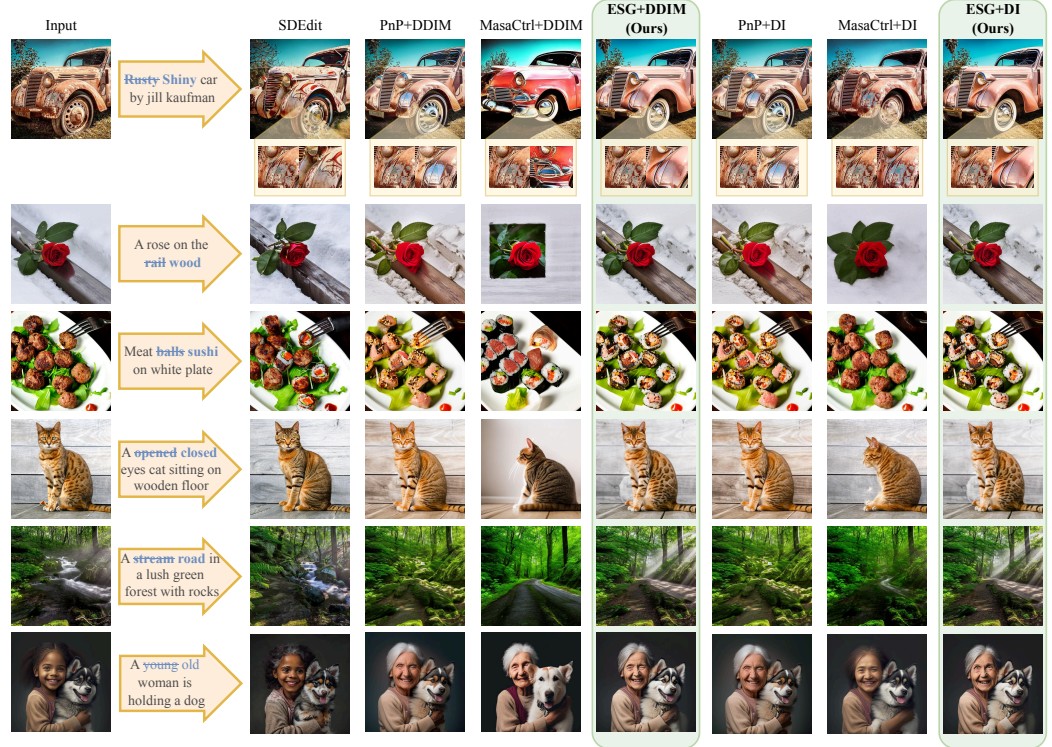

Figure 3: Representative edited images on PIE-Bench. From left to right: input image and target prompt, followed by outputs from SDEdit, PnP + DDIM, MasaCtrl + DDIM, our method + DDIM, PnP + DI, MasaCtrl + DI, and our method + DI.

## 5 EXPERIMENT

To demonstrate the effectiveness and efficiency of our method, we present experimental results using the techniques described above. We provide quantitative and qualitative comparisons of image-to-image translation our method on the PIE-Bench dataset. We also analyze computational performance and conduct an ablation study.

### 5.1 EXPERIMENTAL SETUP

**Datasets.** Our experiments use the PIE-Bench Ju et al. (2023) dataset, which contains 700 images from diverse natural and synthetic scenes, including animals, people, indoor spaces, outdoor views, and computer-generated illustrations and this dataset includes ten categories of image modifications, following the guidelines in Ju et al. (2023). The appendix includes more dataset's details.

**Evaluation Metrics.** Following the evaluation framework in Ju et al. (2023), we use eight metrics grouped into four categories to assess visual quality and efficiency:

- **Structural fidelity:** Measured using DINO-ViT feature similarity Tumanyan et al. (2022) to evaluate high-level semantic correspondence.
- **Background preservation:** Assessed in unmodified regions using PSNR, LPIPS Zhang et al. (2018), MSE, and SSIM Wang et al. (2004).
- **Prompt-image consistency:** Evaluated via CLIP similarity Radford et al. (2021); Wu et al. (2021) between the edited image and the target prompt, computed both globally and within the mask.
- **Inference speed and memory:** Measured as the inference time and GPU memory usage required for image generation.

Table 1: Comparison of different inversion and editing methods on structure, background preservation, and CLIP similarity. We compared our method with SDEdit, MasaCtrl and PNP in DDIM inversion, Null-Text (NT) inversion and Direct inversion (DI).

| Method | | Structure | Background Preservation | | | | CLIP Similarity | |
|---|---|---|---|---|---|---|---|---|
| Editing | Inverse | $DIS_{\times 10^{-3}}\downarrow$ | PSNR ↑ | $LPIPS_{\times 10^{-2}}\downarrow$ | $MSE_{\times 10^{-3}}\downarrow$ | $SSIM_{\times 10^{-2}}\uparrow$ | Whole ↑ | Edited ↑ |
| SDEdit | Other | 31.6 | 21.7 | 14.9 | 9.9 | 72.1 | 31.1 | 26.3 |
| MasaCtrl | DDIM | 75.8 | 17.4 | 21.8 | 24.3 | 70.8 | 32.7 | 28.4 |
| PnP | DDIM | 32.0 | 21.4 | 12.6 | 9.8 | 78.5 | 31.0 | 27.2 |
| **Ours** | DDIM | **26.7** | **22.2** | **11.5** | **8.8** | **79.9** | 30.8 | 27.1 |
| MasaCtrl | NT | 60.7 | 18.3 | 18.7 | 20.0 | 73.3 | 32.5 | 28.2 |
| PnP | NT | 30.1 | 22.7 | 12.1 | 7.6 | 79.0 | 30.1 | 26.5 |
| **Ours** | NT | **30.0** | **23.2** | **11.2** | **7.2** | **80.1** | 30.0 | 26.3 |
| MasaCtrl | DI | 34.8 | 21.1 | 11.9 | 11.8 | 79.6 | 30.1 | 26.2 |
| PnP | DI | 30.5 | 21.5 | 12.2 | 9.7 | 78.6 | 31.0 | 27.2 |
| **Ours** | DI | **26.6** | **22.3** | **11.5** | **8.8** | **80.0** | 30.8 | 27.1 |
| ControlNet | DDIM | 57.9 | 18.1 | 17.1 | 20.7 | 72.9 | 31.3 | 27.5 |
| **+ Ours** | DDIM | **27.9** | **21.5** | **10.9** | **10.2** | **79.7** | 30.8 | 27.0 |
| ControlNet | NT | 52.6 | 19.8 | 15.4 | 14.6 | 75.6 | 31.1 | 27.0 |
| **+ Ours** | NT | **23.2** | **23.9** | **9.4** | **6.2** | **82.3** | 30.7 | 26.8 |
| ControlNet | DI | 55.7 | 18.2 | 16.9 | 19.9 | 73.4 | 30.6 | 26.9 |
| **+ Ours** | DI | **21.7** | **22.2** | **6.7** | **6.1** | **81.1** | 30.1 | **27.4** |

**Baselines.** To evaluate our method's performance in text-guided image editing, we compare it with several inversion and editing combinations. For inversion method, we employ three techniques: standard DDIM inversion Song et al. (2021), Null-text (NT) inversion Mokady et al. (2023), and Direct Inversion (DI) Ju et al. (2023). For editing, we employ Stochastic Differential Editing (SDEdit) Meng et al. (2021), ControlNet Zhang et al. (2023a) conditioned on depth maps, Plug-and-Play (PnP) diffusion Tumanyan et al. (2023), and Mutual Self-Attention Control (MasaCtrl) Cao et al. (2023).

**Configurations.** We implement ControlNet with depth conditioning on Stable Diffusion v2.0. All other methods use conditional Latent Diffusion Models (LDMs) Rombach et al. (2022) based on Stable Diffusion v2.1 to ensure a consistent framework for fair comparison. In all experiments, we use DDIM sampling with 50 steps and set the guidance scale $\omega = 7.5$ for every method. We use the same feature and attention map layers, as well as the same value of $\tau$, for the pre-trained model; details are provided in the Appendix.

## 5.2 OVERALL PERFORMANCE

**Qualitative Evaluation.** Figure 3 presents results from various training-free methods for text-driven image-to-image translation on PIE-Bench. From left to right, the columns display the input image, followed by edits from SDEdit, PnP + DDIM, MasaCtrl + DDIM, our method + DDIM, PnP + DI, MasaCtrl + DI, and our method + DI. Figure 3 demonstrates that our method more effectively preserves the spatial structure of the central object and its surrounding background, all while achieving the intended semantic alterations. For instance, it maintains the dents on the shiny cars in the first row, the rose's shape with light white snow in the second row, the closed-eyed cat with spots against the same-colored wall behind (unlike striped cats in other methods) in the fourth row, and the shine on the road in the last row. Moreover, our method succeeds in semantic translations where other methods fail, such as changing meatballs to meat sushi in the second row. Overall, our approach balances shape preservation and semantic transfer effectively.

**Quantitative Evaluation.** Table 1 presents quantitative results comparing our method with baselines on structure similarity, background preservation (PSNR, LPIPS, MSE, SSIM), and CLIP similarity (whole and edited). Our method outperforms competing techniques, achieving substantial improvements in structure and background retention while maintaining comparable CLIP similarity scores. For instance, our method lowers the structure distance from $32.0 \times 10^{-3}$ (PnP + DDIM) to $26.7 \times 10^{-3}$ (Ours + DDIM). Additionally, integrating our approach with other null-text or direct inversion strategies, such as NT or DI, yields further gains in structure metrics and background fidelity.

Table 2: Ablation study of our method + DDIM on PIE-Bench, evaluating structure preservation, background retention, and CLIP similarity. Rows represent variants: classifier-free guidance without or with unconditional branch (CFG w/o or w/ Uncon), with Target-Guided (TG) injection, and with TG + Source Noise Shared (SNS).

| Method | Structure | Background Preservation | | | | CLIP Similarity | |
|---|---|---|---|---|---|---|---|
| Editing | $\text{Dis}_{\times 10^{-3}} \downarrow$ | PSNR ↑ | $\text{LPIPS}_{\times 10^{-2}} \downarrow$ | $\text{MSE}_{\times 10^{-3}} \downarrow$ | $\text{SSIM}_{\times 10^{-2}} \uparrow$ | Whole ↑ | Edited ↑ |
| CFG W/O Uncon | 21.9 | 24.9 | 9.1 | 5.1 | 82.6 | 29.4 | 25.8 |
| CFG W/ Uncon | 149.4 | 14.4 | 28.6 | 50.0 | 64.3 | 29.6 | 26.0 |
| + TG | 60.3 | 18.1 | 19.2 | 20.8 | 72.0 | 31.8 | 28.0 |
| + TG + SNS | 26.7 | 22.2 | 11.5 | 8.8 | 79.9 | 30.8 | 27.1 |

This demonstrates that our framework acts as a modular upgrade for established works. Integrating our approach with other inversion strategies, such as NT or DI, further enhances structure metrics and background fidelity. This demonstrates that our framework serves as a modular upgrade for existing methods. For example, Ours+DI reduces structure distance from $34.8 \times 10^{-3}$ (MasaCtrl+DI) to $26.6 \times 10^{-3}$, with PSNR increasing to 22.3 and LPIPS dropping to $11.5 \times 10^{-2}$. Moreover, our method integrates seamlessly into different diffusion models, such as ControlNet. Compared to ControlNet + DI without our method, adding ours improves structure (from $55.7 \times 10^{-3}$ to $21.7 \times 10^{-3}$) and CLIP similarity in the edited region (from 26.9 to 27.4). These gains highlight our method's ability to balance structural preservation with text prompt adherence.

We also compare time and GPU memory usage for MasaCtrl, PnP, and our method when generating one edited image from latent noise on an RTX 4090 in Table 3. Compared with MasaCtrl and PnP, our method decrease approximately 63% and 43% of the time and 35% and 26% of the GPU memory, respectively. Relative to PnP, our method requires about 57% of the time and 84% of the GPU memory. This illustrates the efficiency of our method compared to others that require an additional auxiliary branch.

Table 3: Computational performance comparison among MasaCtrl, PnP, and our method. Bar graphs illustrate processing time cost and GPU memory usage for generating one image on an RTX 4090.

| Method | Memory (GB) | Times (S) |
|---|---|---|
| MasaCtrl | 9.1 | 6.7 |
| PnP | 7 | 4.4 |
| **Ours** | **5.9** | **2.5** |

## 5.3 ABLATION STUDY

Table 2 evaluates the components of our method with DDIM inversion, comparing classifier-free guidance without or with the unconditional branch, with Target-Guided (TG) injection, and with TG + Source Noise Shared (SNS). The first row shows that editing with only the conditional branch preserves structure but loses semantic alignment with the target prompt, as discussed in Section 1. The second row, using standard classifier-free guidance (vanilla baseline), severely deforms the original image, leading to high structure distance. The third row, incorporating the Target-Guided unconditional branch, produces shapes closer to the input image compared to standard classifier-free guidance (the structure distance decrease from $149.4 \times 10^{-3}$ to $60.3 \times 10^{-3}$). The last row, applying our full method (Target-Guided and Source Noise Shared unconditional branch), further improves shape fidelity compared to TG alone (the structure distance decrease from $60.3 \times 10^{-3}$ to $26.7 \times 10^{-3}$). These results confirm our method's improved balance between structural preservation and text prompt adherence.

## 6 CONCLUSION

In this work, we addressed structural drift in text-guided image editing with CFG. Our empirical analysis uncovered the trade-off between semantic alignment and structural fidelity, identifying the unconditional branch as the primary culprit. We introduced Efficient Self-Guided Editing (ESG), a framework that reuses features from the target branch and shares source noise to enhance coherence in the unconditional branch without incurring additional computational costs. Experiments on PIE-Bench demonstrate that ESG outperforms baselines in structure and background preservation while achieving comparable semantic alignment, all with reduced inference time and GPU memory usage.

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

# Supplementary Material

## A SUPPLEMENTARY EXPERIMENTS

In this section, we provide additional dataset details and ablation studies for hyperparameters, including quantitative evaluations.

### A.1 SUPPLEMENTARY DATASET DETAILS

Each image in PIE-Bench has five associated labels: an initial prompt describing the source image, a target prompt specifying the desired change, an edit directive, a textual description of the edited object, and a binary mask indicating the region to modify. The initial prompts are generated automatically using BLIP-2 Li et al. (2023). In contrast, the target prompts and directives are produced by GPT-4 Achiam et al. (2023) and then manually refined for accuracy and consistency.

### A.2 SUPPLEMENTARY ABLATION STUDY

In this subsection, we discuss the impact of self-attention layer injection in the target-guided unconditional branch and the threshold $\tau$ in source noise sharing. Here, we focus on self-attention layers, as $\tau$ is analyzed separately. Each experiment runs three time and get the average result.

**Self-Attention Layers.** We use the same feature layer as PnP Tumanyan et al. (2023) ($l = 4$) from the conditional branch for injection into the unconditional branch. We examine how varying the number of self-attention layers affects structure preservation and semantic similarity. The U-Net has 6 groups of self-attention layers in the down and up blocks. We evaluate by incrementally adding these groups, starting from those closest to the output. Results with $\tau = 20$ are shown in Table 4 for ESG + DDIM on PIE-Bench. As shown, injecting more self-attention layers increases structural similarity to the input image but decreases semantic alignment with the target prompt. Thus, we select 5 groups to balance structural preservation and text prompt adherence.

Table 4: Ablation study of our method + DDIM with different numbers of self-attention layers injected on PIE-Bench, evaluating structure preservation, background retention, and CLIP similarity.

| Self-Attn | Structure | Background Preservation | | | | CLIP Similarity | |
|---|---|---|---|---|---|---|---|
| Layers | $\text{Dis}_{\times 10^{-3}} \downarrow$ | PSNR $\uparrow$ | $\text{LPIPS}_{\times 10^{-2}} \downarrow$ | $\text{MSE}_{\times 10^{-3}} \downarrow$ | $\text{SSIM}_{\times 10^{-2}} \uparrow$ | Whole $\uparrow$ | Edited $\uparrow$ |
| 0 | 128.7 | 15.2 | 26.7 | 42.1 | 66.1 | 30.1 | 26.4 |
| 1 | 57.7 | 19.1 | 18.0 | 18.4 | 74.1 | 31.1 | 27.4 |
| 2 | 35.8 | 21.2 | 13.6 | 11.4 | 77.9 | 30.8 | 27.2 |
| 3 | 33.4 | 21.4 | 13.0 | 11.0 | 78.5 | 30.8 | 27.2 |
| 4 | 31.7 | 21.6 | 12.5 | 10.4 | 78.8 | 30.7 | 27.1 |
| **5** | 26.7 | 22.2 | 11.5 | 8.8 | 79.9 | 30.8 | 27.1 |
| 6 | 23.4 | 22.8 | 10.7 | 7.7 | 80.7 | 30.6 | 27.0 |

Table 5: Ablation study of our method + DDIM with different values of $\tau$ on PIE-Bench, evaluating structure preservation, background retention, and CLIP similarity.

| Self-Attn | Structure | Background Preservation | | | | CLIP Similarity | |
|---|---|---|---|---|---|---|---|
| $\tau$ | $\text{Dis}_{\times 10^{-3}} \downarrow$ | PSNR $\uparrow$ | $\text{LPIPS}_{\times 10^{-2}} \downarrow$ | $\text{MSE}_{\times 10^{-3}} \downarrow$ | $\text{SSIM}_{\times 10^{-2}} \uparrow$ | Whole $\uparrow$ | Edited $\uparrow$ |
| 0 | 60.3 | 18.1 | 19.2 | 20.8 | 72.0 | 31.8 | 28.0 |
| 5 | 50.0 | 19.1 | 17.1 | 16.5 | 74.2 | 31.6 | 27.8 |
| 10 | 40.8 | 20.3 | 15.0 | 13.1 | 76.4 | 31.3 | 27.6 |
| 15 | 32.3 | 21.4 | 12.9 | 10.4 | 78.5 | 31.0 | 2.3 |
| **20** | 26.7 | 22.2 | 11.5 | 8.8 | 79.9 | 30.8 | 27.1 |
| 25 | 24.0 | 22.8 | 10.6 | 7.8 | 80.8 | 30.4 | 26.8 |

**Source Noise Shared** $\tau$. We evaluate how different values of $\tau$ affect structural preservation and text prompt adherence. We use our method + DDIM with 5 groups of self-attention layers and $l = 4$ for feature injection in the following evaluations. As illustrated in Table 5, structural similarity increases and semantic alignment with the target prompt decreases as $\tau$ increases. Thus, we select $\tau = 20$ to balance structural preservation and text prompt adherence.

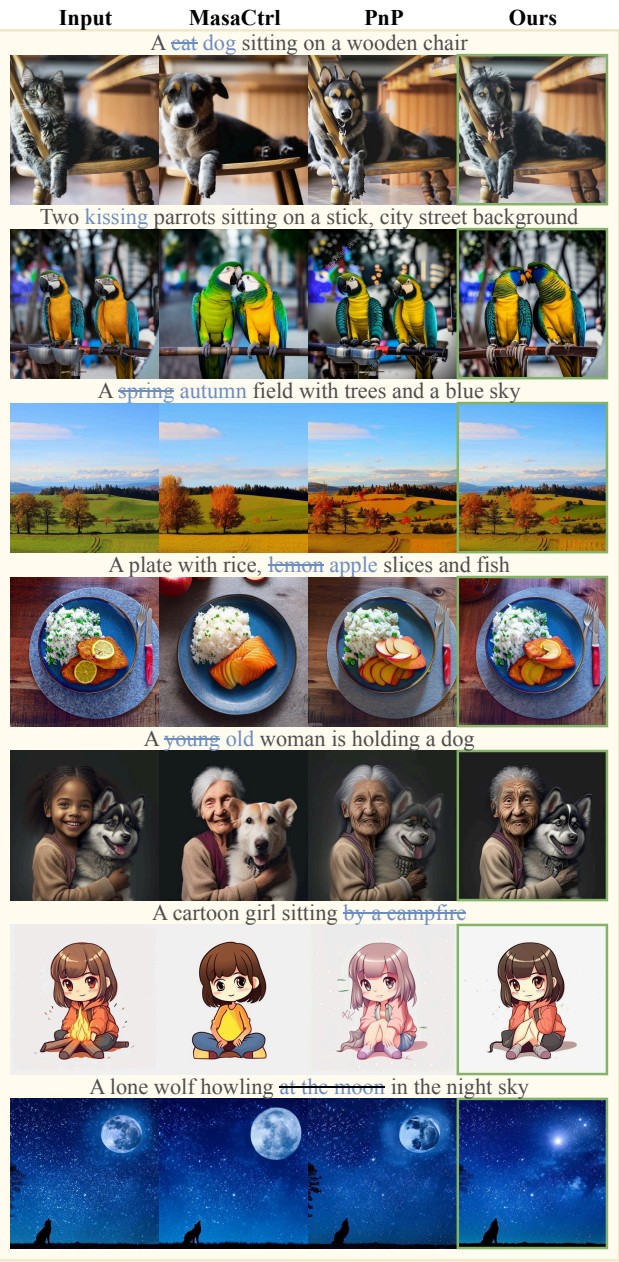

Figure 4: Representative edited images on PIE-Bench with Null-Text Inversion. From left to right: input image, followed by outputs from MasaCtrl, PnP, and our method (green box).

### A.3 SUPPLEMENTARY QUALITATIVE EVALUATIONS

Figure 4 compares images generated by different methods using Null-Text Inversion. From left to right, the images are the input, MasaCtrl, PnP, and ours. This demonstrates that ESG (our method)

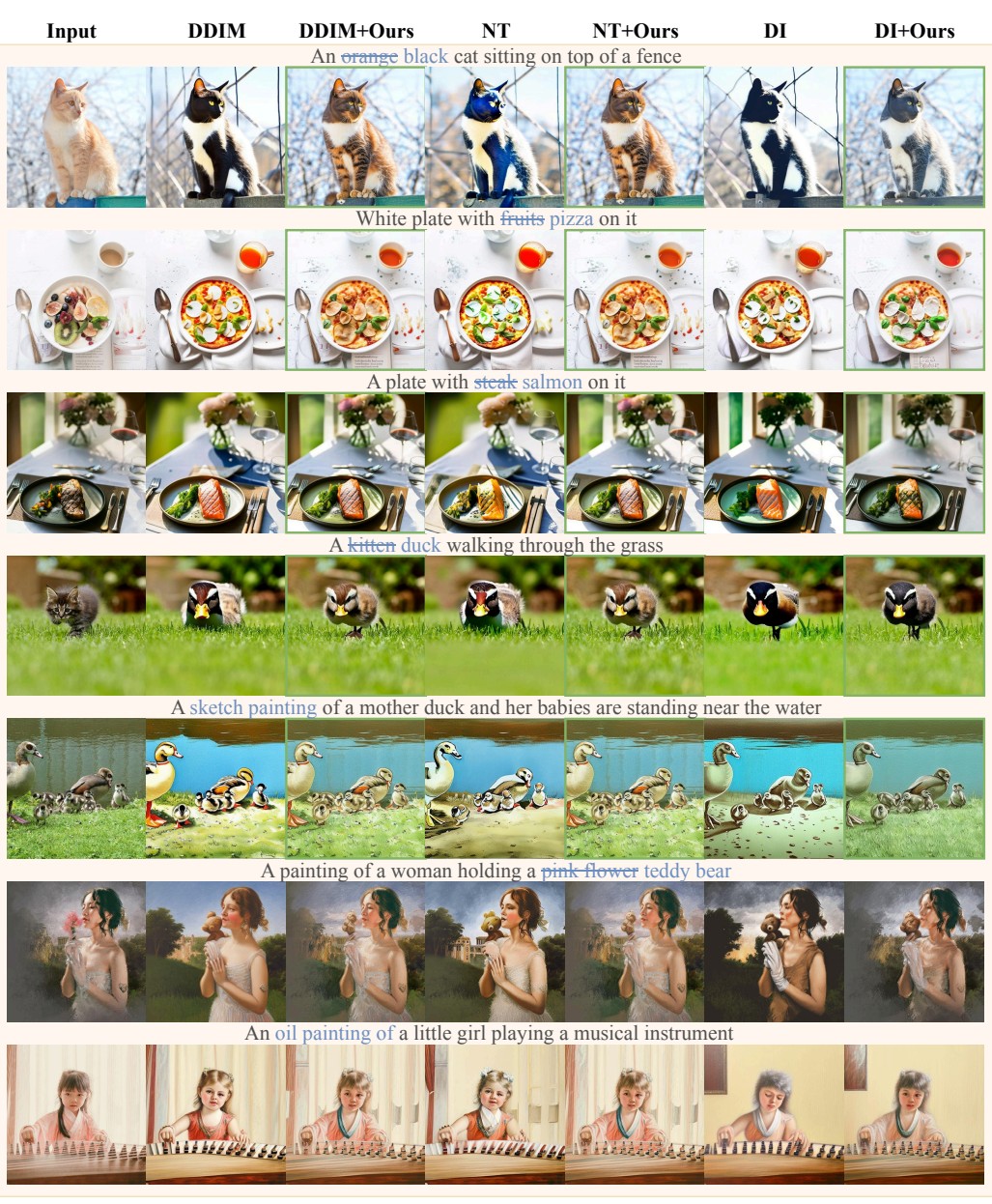

Figure 5: Representative edited images on PIE-Bench. From left to right: input image and target prompt, followed by outputs from ControlNet + DDIM, ControlNet + Ours + DDIM (green box), ControlNet + NT, ControlNet + Ours + NT (green box), ControlNet + DI, ControlNet + Ours + DI (green box).

better preserves structure and background. For example, ESG maintains the fur color consistent with the input image (1st row), the same shape and number of trees (3rd row), edits only the lemon to apple while keeping the background fish (4th row), and removes the moon from the input image (6th row). Additionally, Figure 5 compares images generated with and without our method using different inversion techniques. With our method, the output images achieve target semantics while retaining the input image's structure. For instance, it preserves the words in the background book (2nd row), the red wine in the glass (3rd row), and the woman's appearance and dress (6th row). These figures highlight our method's strong ability to preserve structure and enable effective editing.

## B    THE USE OF LARGE LANGUAGE MODELS (LLMS)

We use some LLMs to polish this paper's writing.

