# OpenReview forum: "Efficient Self-Guided Editing for Text-Driven Image-to-Image Translation"
_ICLR.cc/2026/Conference — ICLR 2026 Conference Withdrawn Submission_

### Official Review · Reviewer_LU8t · 2025-10-30

**Soundness:** 2
**Presentation:** 3
**Contribution:** 2
**Rating:** 4
**Confidence:** 2

**Summary:**

The submissions tackles the problem that in text-guided image editing with diffusion models, classifier-free guidance (CFG) improves semantic alignment but often induces structural drift. The authors identify the unconditional (null-text) branch in CFG as the main source of this structural drift. To mitigate the drift, the submission proposes to inject the mid-level features and self-attention from the target-prompt (conditional) branch to the unconditional branch.

**Strengths:**

The main idea of using the artefacts from the conditional branch for the unconditional one looks novel for me, it does eliminate the need for an extra reconstruction branch (as in PnP/MasaCtrl).

**Weaknesses:**

(1) In terms of experimental evaluation, I did not find the user study comparison, all the quantitative results are reported in terms of automated performance metrics. In my opinion, human-based evaluation is necessary for such kind of papers.

(2) The submission does not compare to the relevant baselines of fast inversion-based editing (LEDITS++: Limitless Image Editing using Text-to-Image Models. Brack et al, An Edit Friendly DDPM Noise Space: Inversion and Manipulations. Huberman-Spiegelglas et al) both in terms of quality and efficiency.

(3) The submission provides quite limited justification for the main design choices (TG, SNS). In my opinion, the authors do not convincingly explain why the target-guidance and the source-noise sharing should jointly improve structure preservation while maintaining edit strength. I appreciate the sections 4.1–4.2 but they are not convincing enough.

**Questions:**

Please, address my concerns in the Weaknesses section.

---

### Official Review · Reviewer_kVTE · 2025-10-30

**Soundness:** 2
**Presentation:** 2
**Contribution:** 2
**Rating:** 2
**Confidence:** 4

**Summary:**

The paper proposes Efficient Self-Guided Editing (ESG) which incorporates two techniques for inversion-based editing aimed at improving unconditional predictions within CFG for higher spatial consistency. Also, by eliminating the need in additional source preservation branch, ESG yields faster editing and lower memory usage. ESG is evaluated on PIE-bench and shows editing improvements over several inversion methods.

**Strengths:**

* The method avoids using the external source-reconstruction branch and thus enables more efficient inversion-based editing.

**Weaknesses:**

* Lack of proper justification for why TG and SNS are reasonable. It remains unclear to me why TG and SNS should improve source preservation and provide strong editing results. I can see some discussions in Secs. 4.1 and 4.2, but they did not help me much. I highly encourage including additional analysis, clarifications, and discussions.

* A few simple baselines that may improve source preservation for inversion-based editing are missing. I believe both are valuable to illustrate the importance of TG and SNS:

1) Interval guidance [1,2], where CFG is turned off for high-noise steps. It should significantly improve spatial consistency while preserving editing strength.

2) Early-stopped DDIM inversion: stopping at intermediate timesteps, e.g., t = 700–800, which also improves source preservation.

* The method is essentially a CFG modification that resembles some improved CFG methods [3,4], but is applied to inversion-based editing. These methods also modify the unconditional prediction, and thus comparisons with them are important. Moreover, other CFG modifications based on prediction projection could also be relevant baselines [5,6].

* Missing baselines on efficient inversion-based image editing [7,8].

* The method is designed for UNet-based diffusion models, while current s.o.t.a. models are based on transformers and flow matching.
Since multiple inversion-based editing methods already exist for the models such as FLUX [9,10,11], I believe it is important to adapt the method to transformer-based models, evaluate it on SOTA models (e.g., FLUX, Cosmos-Predict2) and compare against the recent baselines.

Moreover, among the UNet-based models, the experiments are only on SD2.1, which is now largely outdated. I would expect results on SDXL at least. The comparison with [9] would be valuable as well.

**Minor**

* The experiment in the Introduction is confusing, as it uses specific terminology before any background or literature review is provided. Even well-informed readers may be misled by different variants of “conditional” and “unconditional” branches without a precise description of how these options work.
* Missing brackets for citations: \cite -> \citep

---

[1] Applying Guidance in a Limited Interval Improves Sample and Distribution Quality in Diffusion Models. NeurIPS2025

[2] Invertible Consistency Distillation for Text-Guided Image Editing in Around 7 Steps. NeurIPS2025

[3] CFG++: Manifold-constrained Classifier Free Guidance for Diffusion Models. ICLR2025

[4] Spatiotemporal Skip Guidance for Enhanced Video Diffusion Sampling. CVPR2025

[5] Eliminating Oversaturation and Artifacts of High Guidance Scales in Diffusion Models. ICLR2025

[6] CFG-Zero*: Improved Classifier-Free Guidance for Flow Matching Models. 2025

[7] An Edit Friendly DDPM Noise Space: Inversion and Manipulations. CVPR2024

[8] LEDITS++: Limitless Image Editing using Text-to-Image Models. CVPR2024

[9] Tight Inversion: Image Conditioned Inversion for Real Image Editing. CVPR2025

[10] Semantic Image Inversion and Editing using Stochastic Rectified Differential Equations. ICLR2025

[11] Stable Flow: Vital Layers for Training-Free Image Editing. CVPR2025

**Questions:**

Please address the concerns in Weaknesses

---

### Official Review · Reviewer_zhU6 · 2025-10-31

**Soundness:** 2
**Presentation:** 2
**Contribution:** 1
**Rating:** 2
**Confidence:** 5

**Summary:**

In text-guided image editing, CFG can enhance semantic fidelity but exhibit distortions in the original image’s geometric structure and spatial coherence. To explore the trade-off between structural fidelity and semantic alignment, the authors attempt to perform two systematic empirical studies:
- Compare image editing with and without CFG.
- Compare two distinct prompt-injection strategies, one is injecting a source’s branch into both condition and unconditional branches, and another is injecting only the conditional branch into the unconditional branch.

They further propose Efficient Self-Guided Editing (ESG), a framework designed to enhance structural preservation and semantic fidelity in DDIM-based text-driven image editing, without the need for an additional source reconstruction branch.

**Strengths:**

- Extensive experiments are conducted
- The writting is easy to follow

**Weaknesses:**

- The motivation is not very clear, as the inherent CFG mechanism already controls the trade-off between unconditional and conditional inputs.
- Figure 1 lacks clarity and is not easy to interpret. The text annotations within the figure are difficult to read, which weakens the presentation of the paper’s motivation.
- What does the source’s branch refer to in line 83?
- Figure 2, which presents the main architecture of the method, is not intuitive and is difficult to understand. Both the figure and its caption lack clarity, making it hard for readers to grasp the overall workflow.
- In Figure 3, the visual results of the proposed ESG method appear quite similar to those of PnP, showing only marginal improvements.
- In Table 1, the proposed method performs worse than PnP in terms of CLIP similarity. The paper does not provide an explanation or analysis for this performance gap
- In Table 2, the authors claim that editing with only the condition branch losses semantic alignment with the target prompt. These metrics cannot measure this property. Given visual comparison can help verify this anlaysis.

**Questions:**

- What does the empty set in equation4-7 mean?
- The CGF is used to blend the uncondition and uncontion branches, and it is an ajustable constant. In this paper, the authors do not specify the exact value of this constant.

**Details Of Ethics Concerns:**

The paper mainly lacks visual evidence to support the claims, as well as an analysis of the CFG value settings and their relationship to the proposed method.

---

### Official Review · Reviewer_vrJW · 2025-11-01

**Soundness:** 3
**Presentation:** 2
**Contribution:** 3
**Rating:** 4
**Confidence:** 4

**Summary:**

This paper “Efficient Self-Guided Editing for Text-Driven Image-to-Image Translation” introduces a training-free diffusion editing framework called ESG that aims to balance semantic alignment to the target prompt and preservation of the original image structure. The authors identify that classifier-free guidance often causes structural drift, mainly due to the unconditional (null-text) branch, which tends to diverge spatially during denoising. To fix this, the authors propose two key strategies: a) Target-Guided Unconditional Branch and b) Source Noise Sharing. This approach avoids the extra reconstruction branch needed in methods like MasaCtrl or PnP (prior works), reducing GPU memory and computational cost. Experiments on PIE-Bench show ESG better preserves structure and background while maintaining similar CLIP-based semantic consistency.

**Strengths:**

S1. The paper clearly recognizes the motivation for method design, and it is well-explained in the Introduction section (Sec. 1.)

S2. The paper proposes two novel components for image-to-image translation, which derives critical performance improvements as explained in the Experiment Section.

S3. The proposed method shows superior performance in a quantitative manner compared to baselines (SDEdit, MasaCtrl, and Plug-and-Play).

**Weaknesses:**

**Major Weakness**

W1. This paper only compares with the three baselines; that is, SDEdit, MasaCtrl, and Plug-and-play. However, there are more recent state-of-the-art work for diffusion-based image-to-image translation, such as Diffusion Self-Guidance, LocInv, DDS, Motion Guidance, and etc. I think comparison with more recent works is also required.

W2. The experiment section only focuses on the object change tasks, such as Rusty car → Shiny car and young woman → old woman. I was wondering if the method is applicable to more difficult tasks, such as 1) object addition (or object duplication), 2) object deletion, and 3) enlarging or shrinking the object size. Extensive experiments with additional tasks is required.

W3. The authors use Stable diffusion v2.0 and v2.1 for experiments. However, recent works use the Transformer-based diffusion model, beyond U-Net based Stable Diffusion. Is it possible to apply the proposed method into the DiT-based model, such as Stable Diffusion v3?



**Minor Weakness**

W4. The overall writing can be improved. Especially citations styles (Author et al. (year)) can be changed to improve the readability of the main paper.

**Questions:**

Please check the weakness section.

---

### Note · Authors · 2025-11-19

I have read and agree with the venue's withdrawal policy on behalf of myself and my co-authors.